# VIDEO-STR: REINFORCING MLLMS IN VIDEO SPATIO-TEMPORAL REASONING WITH RELATION GRAPH

## ABSTRACT

Despite the rapid progress of Multimodal Large Language Models (MLLMs) in high-level semantic understanding, they still exhibit notable limitations in fine-grained spatio-temporal reasoning. Existing spatio-temporal methods primarily focus on the video itself, while overlooking the physical information within the video, such as multi-object layouts and motion. Such limitations restrict the use of MLLMs in downstream applications that demand high precision, including embodied intelligence and VR. To address this issue, we present **Video-STR**, a novel graph-based reinforcement method for precise **Video S**patio-**T**emporal **R**easoning. Building upon the capacity of Reinforcement Learning with Verifiable Reward (RLVR) to improve model abilities, we introduce a reasoning mechanism using graph-based Group Relative Policy Optimization (GRPO) method to guide the model in inferring the underlying spatio-temporal topology of scenarios during the thinking process. To resolve the lack of spatio-temporal training data, we construct the **STV-205k** dataset with 205k question-answering pairs, covering dynamic multi-object scenes in both indoor and outdoor environments, to support the model training. Experiments show that Video-STR achieves state-of-the-art results on various benchmarks, outperforming the base model by 13% on STI-Bench, and demonstrating the effectiveness of our approach and dataset. Code, model, and data will be released.

## 1 INTRODUCTION

Multi-modal Large Language Models (MLLMs) have made significant progress in multi-modal understanding and reasoning (Zhao et al., 2025a; Huang et al., 2023; Ye et al., 2024; Liu et al., 2024a; Xu et al., 2025), but still struggle with precise spatio-temporal reasoning in videos (Li et al., 2025c; Cheng et al., 2025; Zheng et al., 2025). Spatio-temporal reasoning entails the inference of spatial and temporal relationships, where the former mainly captures positional topology, and the latter reflects dynamic changes. This requires the model not only to understand the objects' layouts, but also to model their motion. Some works have explored spatio-temporal understanding of videos (Yuan et al., 2025; Wang et al., 2025a; Li et al., 2025a; Liu et al., 2024a), but they mainly focus on changes within the image, overlooking the physical information behind the videos, the exact spatial positions of objects, their trajectories, and the interactions between multiple entities over time, etc.

Reinforcement Learning with Verifiable Reward (RLVR) has been demonstrated as a promising solution to unlock LLM reasoning gains through task-specific reward design (Guo et al., 2025; Yang et al., 2025b). Several works extend it on MLLMs to adapt for video spatio-temporal reasoning (Feng et al., 2025; Li et al., 2025b), which primarily employ two types of reward schemes: video pixel-level localization (Cheng et al., 2025; Wang et al., 2025a; Shi et al., 2024) and 2D cognitive map (Ouyang et al., 2025; Yang et al., 2025a). As shown in Fig. 1 (a), pixel localization is limited to identifying positions in the image plane, without the ability to infer object layouts or distributions in the physical space (Li et al., 2025c). 2D cognitive map depicts the scene with a planar grid map, changes in camera viewpoints and non-rotation-invariant object coordinates can lead to errors in estimating object distributions. To overcome these limitations, we design a graph-based representation for multiple objects, in which nodes correspond to objects and edges characterize the relationships between them. This brings two benefits: (1) beyond localizing individual objects,

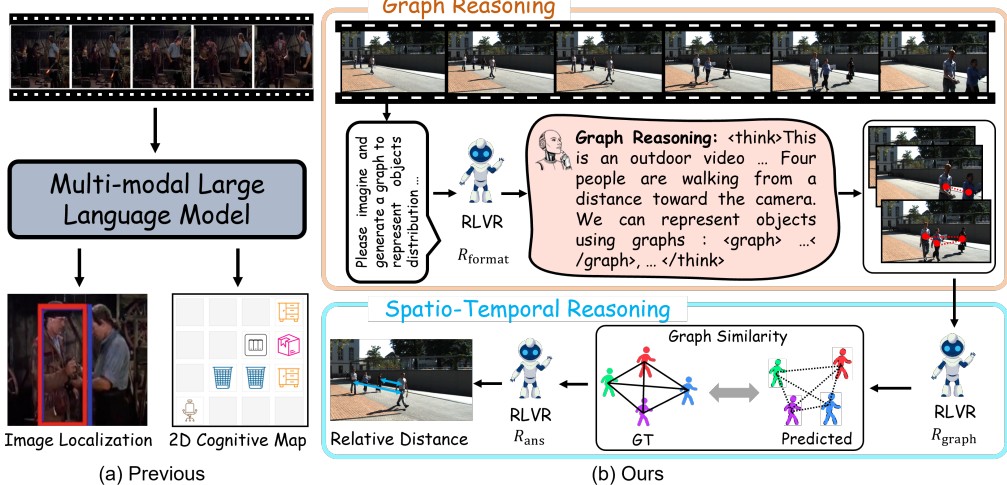

Figure 1: Overview of Video-STR and previous methods. (a) Previous methods localized using images and 2D cognitive maps with spatial cues. (b) Our method, Video-STR, uses graph-based reasoning and RL to infer spatio-temporal relations and object states in videos.

it further effectively characterizes inter-object relationships; (2) unlike object–camera relationships that are affected by dynamic ego viewpoint, inter-object relationships are insensitive to such dynamics, making the graph structure rotation-invariant and robust to video spatio-temporal reasoning.

In this work, we present **Video-STR**, a novel RLVR framework to improve the spatio-temporal reasoning capabilities of MLLMs. Specifically, we extend the Group Relative Policy Optimization (GRPO) (Shao et al., 2024) with a graph reasoning mechanism that enables the model to understand the spatial topology using the inter-object relation graph, where the graph is constructed based on the objects' attributes. To support training, we introduce the STV-205k, a dataset comprising question-answering (QA) pairs derived from the TAO (Dave et al., 2020), KITTI (Geiger et al., 2013) and ScanNet (Dai et al., 2017) datasets. The constructed dataset covers spatial reasoning tasks (e.g., relative distance, object sizes) and temporal reasoning tasks (e.g., appearance order and displacement). Meanwhile, diverse verifiable rewards $R_{ans}$ is introduced to adapt to various QA types in training.

Experiments on STI-Bench (Li et al., 2025c), V-STaR (Cheng et al., 2025), VSI-Bench (Yang et al., 2025a), SPAR-Bench (Zhang et al., 2025b), Video-MME (Fu et al., 2025), and TempCompass (Liu et al., 2024b) demonstrate that our method achieves strong performance in spatial, temporal, and spatio-temporal reasoning. In summary, our main contributions include:

- We construct the STV-205k dataset from TAO, KITTI, and ScanNet, providing 205k diverse QA pairs covering both static spatial and dynamic temporal reasoning, addressing the lack of video spatio-temporal datasets.
- We first employ an inter-object relation graph to provide a more comprehensive characterization of multi-object scenes. Based on this relation graph, we extend the GRPO with a graph reasoning mechanism that explicitly supervises the intrinsic topology among multiple objects to reinforce video spatio-temporal reasoning.
- Extensive experiments are conducted across diverse relevant benchmarks, indicating that **Video-STR** achieves the state-of-the-art performance, validating the effectiveness of the STV-205k dataset and our proposed method.

## 2 RELATED WORKS

**Spatio-Temporal Understanding of MLLMs.** Video MLLMs have made significant advancements on semantics understanding (Hariharan, 2025; Fu et al., 2024), while spatio-temporal understanding remains a significant challenge and has inspired recent contributions (Cai et al., 2024; Li et al., 2024; Chen et al., 2024). Existing works mainly focus on the pixel-level spatial reasoning in 2D images. For example, VideoRefer (Yuan et al., 2025) adopts a versatile spatial-temporal object encoder to capture precise regional and sequential representations. VideoExpert (Zhao et al., 2025b) decouples spatial and temporal reasoning, employing two dedicated modules to capture fine-grained

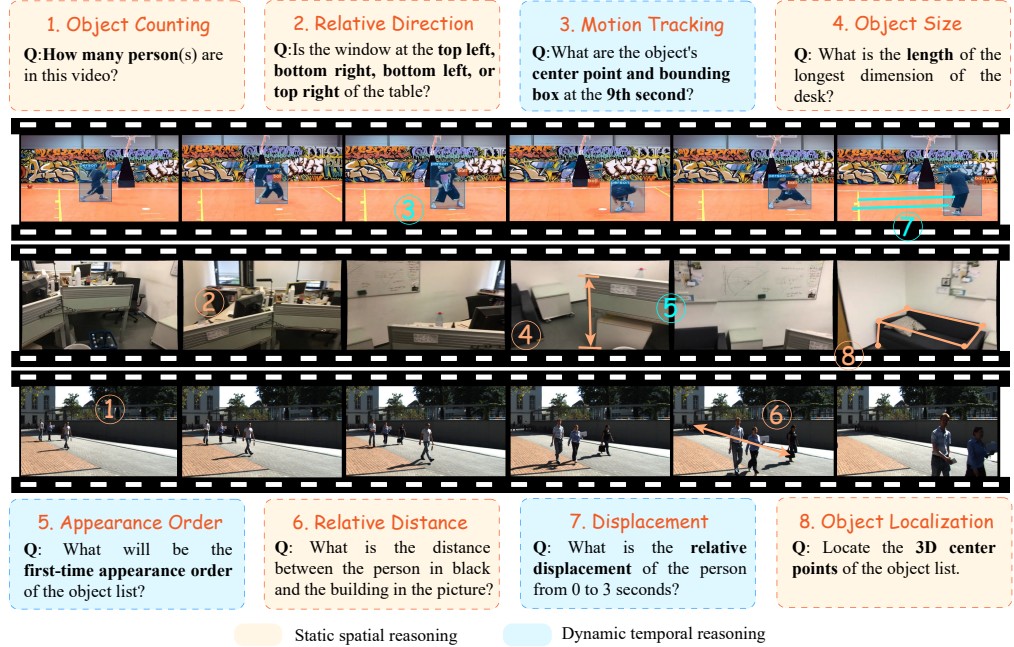

Figure 2: The overview of the constructed dataset.

spatial details and model dynamic temporal patterns, respectively. In contrast to these approaches, our work emphasizes the physical information embedded in videos, thereby enabling more precise spatio-temporal understanding.

**Reinforcement Learning with Verifiable Reward** Recent works, such as OpenAI-o1 (Jaech et al., 2024) and DeepSeek-R1 (Guo et al., 2025), have demonstrated that large models can significantly enhance their reasoning ability through reinforcement learning (RL). Compared with supervised fine-tuning (SFT), it has the advantage of not overfitting (Huang et al., 2025; Da et al., 2025; Shenfeld et al., 2025). In particular, Deepseek-R1 presents the GRPO algorithm, which can bring emergent and robust reasoning with chain-of-thoughts (CoT) in text-based domains. Building on this, Kimi-k1.5 (Team et al., 2025) adopted rule-based RL strategies to improve reasoning across both text and vision modalities. Motivated by this success, several recent efforts have explored extending RL training to MLLMs (Wang et al., 2025b; Zhao et al., 2025c; Xiao et al., 2025). However, applying RL to video understanding remains limited due to challenges in designing temporally sensitive rewards and modeling spatio-temporal consistency. Only a few works, such as SpaceR (Ouyang et al., 2025), have explored this direction, leaving significant space for further research on reward formulation and training efficiency in the video domain.

## 3 DATASET DESIGN AND CONSTRUCTION

### 3.1 DATA COLLECTION

To overcome the scarcity of training data for video spatio-temporal understanding, we develop the STV-205k, a large dataset including 205k QA pairs. To ensure the richness of data sources for a broad spectrum of real-world environments, we collect raw images and videos from three publicly available datasets: TAO (Dave et al., 2020) for motion tracking, ScanNet (Dai et al., 2017) for indoor scene understanding, and KITTI (Geiger et al., 2013) for outdoor scene understanding. The image data primarily supports the model's static context comprehension, encompassing aspects such as spatial reasoning and semantic knowledge. In contrast, video data serves to enhance the model's temporal understanding, including the extraction of sequential dependencies and the inference of motion from visual dynamics. Leveraging the provided frame-by-frame annotations, we transform the sampled data into a standardized format that incorporates object-relevant attributes, such as categories, sizes, and center points, which serve as the ground-truth information for QA tasks.

### 3.2 QA GENERATION

Utilizing the parsed ground truth information of the selected datasets, we automatically generate the QA pairs for spatial-temporal reasoning tasks, which are categorized into multi-choice QA (e.g., rel-

Figure 3: Data statistics of our constructed STV-205k dataset.

ative distance, relative direction, and appearance order) and numerical QA (e.g., object size, absolute distance, and object counting). The QA examples are illustrated in Fig. 2.

- **Object Counting (Image, video).** Given the ground truth labels, we can obtain the number of each type of object in the videos of images.
- **Relative Direction (Image, video).** Based on the centre points of two objects, we can infer the spatial relation of them (e.g., left-up and right-down).
- **Relative Distance (Image, video).** We randomly select a pair of distinct objects from the video or image and embed them into a question template. The Euclidean distance between the centre points of the two objects is then calculated and used as the corresponding answer.
- **Appearance Order (Video).** For the video data, we record the frame index where an object first appears and randomly select a subset of them to generate questions and answers.
- **Object Size (Image, Video).** The size information of objects is obtained from the bounding box annotations in the dataset (e.g., length, width, and height). Besides, for indoor scenes, we also record the area of the indoor space, which is measured in square meters.
- **Motion Tracking (Video).** We use the sequence of bounding boxes to model the motion tracking of the objects in the video, including both 2D and 3D data.
- **Object Localization (Image, video).** The centre point of the object is used as the answer.
- **Displacement (Video).** For the video, we obtain the displacement using the travel distance of specific objects between time stamps.

To ensure the quality of QA pairs and prevent the model from naively memorizing the answers, we rigorously filter the sampled data. For multiple-choice QA, the positions of correct answers are randomly shuffled to balance the answer distribution and mitigate positional bias. Then, we perform synonym substitution on the questions to enhance the model's comprehension ability. Finally, we obtain 205k QA pairs covering diverse tasks.

**Dataset Statistics.** The dataset statistics of STV-205k are illustrated in Fig. 3. According to the task type, a total of 205k samples are divided into two categories: spatial reasoning and temporal reasoning. STV-205k dataset comprises both images and videos and encompasses various verifiable QA types: multi-choice, numerical, and IoU questions. More details for data collection and generation are provided in the Appendix D.

## 4    ST-SPECIFIC RLVR

To enhance video spatial-temporal reasoning in MLLMs, we introduce Video-STR, a reinforcement learning framework that extends GRPO by incorporating a series of verifiable specific reward functions to adapt to the constructed QA pairs. Besides, a novel graph-based reasoning mechanism is introduced into the process of MLLM thinking to guide spatio-temporal reasoning.

### 4.1    VERIFIABLE REWARD FUNCTIONS

To effectively supervise model outputs across diverse QA types, including multiple-choice, numerical, and IoU, we design a set of verifiable reward functions that evaluate responses $\hat{y}$ based on task-specific criteria, assessing either the correctness or the format of the answer.

**Format Reward.** To regulate the responses $\hat{y}$ of the model to conform to a predefined structure, a format reward $R_{format}$ is defined based on whether the model wraps the reasoning process and

answer into `<think>` $\cdots$ `</think>` and `<answer>` $\cdots$ `</answer>` tags:

$$R_{format}(\hat{y}) = \begin{cases} 1, & \text{if } \hat{y} \text{ matches format,} \\ 0, & \text{otherwise.} \end{cases} \quad (1)$$

**Multi-choice Reward.** For multi-choice QA, the reward $R_{mc}$ is binary, depending on whether it matches the ground truth $y$:

$$R_{mc}(\hat{y}, y) = \begin{cases} 1, & \text{if } \hat{y} = y, \\ 0, & \text{otherwise.} \end{cases} \quad (2)$$

**Numerical Reward.** For the numerical QA, we compute the relative accuracy of the predicted results, which is formulated as:

$$R_{num}(\hat{y}, y) = \max(0, 1 - |\hat{y} - y|/y) \quad (3)$$

**IoU Reward.** For the spatio-temporal perception such as object tracking, it requires the MLLM to output the predicted bounding boxes in the video that is associated with the content of a given textual query. Evidently, we can use the Intersection over Union (IoU) between the predicted area by the model $\mathcal{I}_{\hat{y}}$ and the ground-truth area $\mathcal{I}_y$ as the reward function. This reward function effectively characterizes the accuracy of the interval predicted by the model.

$$R_{IoU}(\mathcal{I}_{\hat{y}}, \mathcal{I}_y) = |\mathcal{I}_{\hat{y}} \cap \mathcal{I}_y| / |\mathcal{I}_{\hat{y}} \cup \mathcal{I}_y| \quad (4)$$

### 4.2 GRAPH-BASED SPATIO-TEMPORAL REASONING

The base reward functions lack explicit reward signals to ensure a genuine understanding of spatio-temporal information, and they may merely lead the model to memorize the answers. We introduce a graph-based reasoning mechanism that serves to facilitate spatio-temporal reasoning within the thinking process. As shown in Fig. 2, we adopt the graph-based representation to formulate the topology of multiple objects and their relationships such as object locations, sizes, and the spatial structure of object distribution.

A graph $\mathcal{G}$ is characterized by $(\mathcal{V}, \mathcal{E}, \mathcal{A})$, where $\mathcal{V}$, $\mathcal{E}$ and $\mathcal{A}$ are the set of nodes, edges and attributions. Each edge $e_{ij}$ that connects node $v_i$ and $v_j$ is assigned an attribute $a_{ij} = (d_{ij}, \theta_{ij})$, and node attributes are denoted as $a_{ii} = (x_i)$ for node $v_i$, where $x_i$ is the location, $d_{ij}$ is the Euclidean distance and $\theta_{ij}$ is the angle of relative direction. The model is prompted to generate such a graph to image the distribution of multiple objects within the scene, thereby enhancing the spatial understanding. To assess the quality of the generated graphs, we design a specific reward function to explicitly supervise the accuracy of the model inference. First, we calculate the relative accuracy between the predicted object and the ground truth object by their relative distance:

$$R_n = \sum_{i=1}^{k} \left( \frac{n_i}{\sum_j^k n_j} \times (exp(-||x_i - x_{i'}||_2)) \right) \quad (5)$$

where $k$ is the number of object categories. $n_i$, $n_j$ is the number of $i$-th and $j$-th object. $x_i$ is the predicted center point coordinates of the object, while $x_{i'}$ denotes the ground truth. Besides, for the edges, the reward is formulated as:

$$R_e = \frac{1}{2} \sum_{i=1}^{n} \sum_{j=1}^{n} (exp(-|d_{ij} - d_{i'j'}|) + exp(-|\theta_{ij} - \theta_{i'j'}|)) \quad (6)$$

where $d_{ij}$ denotes the predicted Euclidean distance between $i$-th object and $j$-th object and $d_{i'j'}$ is the corresponding ground truth. $\theta_{ij} \in [0, \pi]$ denotes the angle between $d_{ij}$ and $d_{i'j'}$, serving as a measure of the degree of consistency in objects' relative spatial relationship. Then the graph-based reasoning mechanism of an image is defined as the sum of $R_e$ and $R_n$. For the video, we calculate the mean reward of the selected $k$ frames from the video

$$R_{graph} = \frac{1}{k} \sum_{i}^{k} R_n^{(k)} + R_e^{(k)} \quad (7)$$

To illustrate the reasonableness of the reward function design, we state two theorems, and the proofs are provided in the Appendix A.

**Theorem 1.** Graph-based multi-object topological representations exhibit rotation invariance.

$$R_e(\mathcal{R}X, C) = R_e(X, C) \tag{8}$$

where $\mathcal{R}$ is the rotation, $X$ is the coordinates of the object's location, $C$ is the object category, and $\Phi$ is the representation method.

**Theorem 2.** Rewards based on graph similarity effectively promote the model's comprehension of topological structures.

To further enhance the quality of reasoning, we introduce a length-based reward to regulate the length of the model's output, thereby striking a balance between promoting deeper reasoning and preventing overthinking. Specifically, the model will receive an additional reward $R_l = \omega$ if the output answer is correct and the response length falls within a predefined interval $[l_{min}, l_{max}]$. The total reward can be formulated as:

$$R_{total} = \begin{cases} R_{format} + R_{ans} + R_{graph} + R_l, & \text{if } R_{ans} > 0.8, \\ R_{format} + R_{ans} + R_{graph}, & \text{otherwise.} \end{cases} \tag{9}$$

where $R_{ans}$ denotes the reward for the answer ($R_{mc}$, $R_{num}$ or $R_{IoU}$).

## 4.3 GRPO TRAINING

GRPO is an extension of Proximal Policy Optimization (PPO) (Schulman et al., 2017) that introduces the group of candidates responses, thereby eliminating dependency on the critic model.

Given an input question $q$, GRPO generates candidate responses $o = \{o_1, ..., o_G\}$ through policy sampling, $G$ is the number of output responses. The corresponding reward scores are $\{r_1, ..., r_G\}$. Then GRPO computes their mean and standard deviation for normalization and defines the quality:

$$A_i = (r_i - \text{mean}(\{r_i\}_{i=1}^G)) / \text{std}(\{r_i\}) \tag{10}$$

where $A_i$ denotes the relative quality of the $i$-th answer. The final training objective considers preventing the optimized policy $\pi_\theta$ from deviating far from the original MLLM parameters $\pi_{\text{ref}}$ by adding a KL-divergence term $\mathcal{D}_{KL}(\cdot||\cdot)$, which is formulated as:

$$J(\theta) = E_{q,\{o_i\}}\left[\frac{1}{G}\sum_{i=1}^{G}(\min(\frac{\pi_\theta(o_i|q)}{\pi_{\theta_{\text{old}}}(o_i|q)}A_i, \text{clip}(\frac{\pi_\theta(o_i|q)}{\pi_{\theta_{\text{old}}}(o_i|q)}, 1-\epsilon, 1+\epsilon)A_i)) - \beta\mathcal{D}_{KL}(\pi_\theta||\pi_{\text{ref}})\right] \tag{11}$$

where $\beta$ is a regularization coefficient, preventing excessive deviation from the reference policy during optimization and $\epsilon$ is a positive coefficient that limits the policy updating degree.

## 5 EXPERIMENTS

### 5.1 EXPERIMENTAL SETUP

**Implementation Details.** In the training process, we employ Qwen2.5-VL-7B-Instruct (Bai et al., 2025) as the base model. We use Adam optimizer with a learning rate of 1e-6 to train our model. For each sample, 8 responses are generated to compute the group-based advantage of GRPO. The hyperparameter $\beta$ in the KL-divergence term is set to 0.04. Training is performed with a batch size of 1 on 8 NVIDIA H100 80GB GPUs. To ensure training stability, we apply a weight decay rate of 0.01 and clip the maximum gradient norm to 5. The maximum completion length is set as 1024 tokens. The preferred length range of response is set to $[320, 512]$ and the reward value is 0.2. For efficiency considerations, we limit the maximum number of video frames to 16 during training, where each frame is processed at a resolution of $128 \times 28 \times 28$. During inference, the number of video frames is standardized to 32 and each frame is increased to $448 \times 28 \times 28$ for better accuracy.

**Benchmarks.** We evaluate our model using multiple benchmarks: STI-Bench (Li et al., 2025c) and V-STaR (Cheng et al., 2025) for spatio-temporal reasoning, VSI-Bench (Yang et al., 2025a) and SPAR-Bench for spatial reasoning, and Video-MME (Fu et al., 2025) and TempCompass (Liu et al., 2024b) for temporal reasoning. Various mainstream closed-source and open-source MLLM baselines are used. Details are provided in Appendix C.

Table 1: Results of Video-STR and baselines on various video benchmarks including spatio-temporal benchmarks (V-STaR, STI-Bench), spatial reasoning benchmarks (VSI-Bench, SPAR-Bench) and temporal reasoning benchmarks (Video-MME, TempCompass).

| Model | Params | Spa.-Temp. Reasoning | | Spa. Reasoning | | Temp. Reasoning | |
|---|---|---|---|---|---|---|---|
| | | STI | V-STaR | VSI | SPAR | Video-MME | TC |
| *Mainstream Commercial Models* | | | | | | | |
| GPT-4o (Hurst et al., 2024) | - | 34.8 | 39.5 | 34.0 | 36.4 | 71.9 | 73.8 |
| Gemini-2.0-Flash (Team et al., 2024) | - | 38.7 | – | 45.4 | – | – | – |
| Claude-3.7-Sonnet (Anthropic, 2025) | - | 40.5 | – | – | 21.8 | – | – |
| *Open-Source Models* | | | | | | | |
| Qwen2.5-VL-7B-Instruct (Bai et al., 2025) | 7B | 34.7 | 35.2 | 33.0 | 33.1 | 56.3 | 71.1 |
| MiniCPM-V-2.6 (Yao et al., 2024) | 8B | 26.9 | – | – | – | – | 54.9 |
| VideoLLaMA3-7B (Zhang et al., 2025a) | 7B | 26.9 | 27.0 | – | – | **66.2** | 68.1 |
| Video-R1 (Feng et al., 2025) | 7B | 33.3 | 35.4 | 35.8 | – | 59.3 | **73.2** |
| VideoChat-R1 (Li et al., 2025b) | 7B | 34.3 | 36.5 | 38.0 | – | 58.8 | – |
| SpaceR (Ouyang et al., 2025) | 7B | 35.2 | 34.5 | 45.7 | 37.6 | 57.9 | 71.4 |
| Ours (SFT) | 7B | 35.3 | 34.9 | 41.8 | 33.0 | 57.2 | 69.8 |
| Ours | 7B | **39.3** | **37.8** | **46.5** | **37.9** | 58.1 | 72.1 |

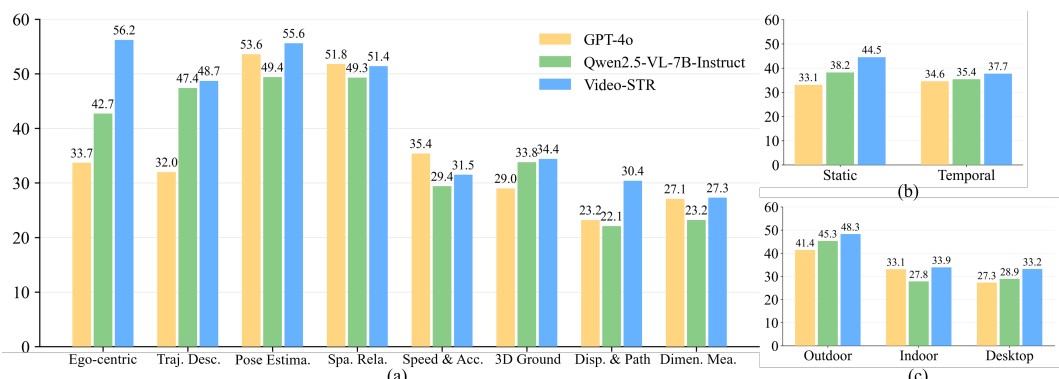

Figure 4: (a) Performance on sub-tasks. (b) Performance on static and dynamic tasks. (c) Performance in different scenarios.

## 5.2 MAIN RESULTS

**Overall Performance.** Table. 1 shows the evaluation results on five relevant benchmarks. We can see that our model outperforms the base model Qwen2.5-VL-7B-Instruct across all benchmarks. Our model surpasses GPT-4o in spatio-temporal reasoning benchmarks, highlighting the capability of spatio-temporal reasoning. In addition, our model also shows improvements in video temporal understanding, indicating that the enhanced spatial understanding did not lead to overfitting, and the model maintained its generalization ability in video comprehension.

**Comparison with SFT.** We also compare the effectiveness of our proposed method with SFT. As shown in Table. 1, While SFT achieves localized improvements on benchmarks including STI-Bench and VSI-Bench, it simultaneously incurs performance degradation on other tasks, reflecting its limited generalizability due to overfitting. In contrast, our method consistently enhances performance across both spatial reasoning and video understanding benchmarks, underscoring its superior generalization capability. These findings substantiate the claim that "RLVR generalizes, whereas SFT memorizes," indicating RLVR as a more effective training paradigm for advancing spatial reasoning in MLLMs. **Performance on sub-tasks.** Fig. 4 shows the results of our proposed model on the sub-tasks of STI-Bench. It can be seen that the model achieves improvements over the base model across all sub-tasks, with particularly notable gains in tasks involving temporal and spatial coupling.

Table 2: Ablation study on different components.

| Method | Frames | Spatio-Temporal | | Spatial Reasoning | | Temporal Reasoning | |
|---|---|---|---|---|---|---|---|
| | | STI-Bench | V-STaR | VSI-Bench | SPAR-Bench | Video-MME | Temp-Compass |
| w/o-spatial subset | 32 | 35.7 | 36.0 | 36.3 | 33.7 | 57.0 | 71.2 |
| w/o-temporal subset | 32 | 36.3 | 34.9 | 44.9 | 35.4 | 56.4 | 71.0 |
| w/o-graph-based reasoning | 32 | 36.9 | 37.1 | 45.8 | 37.4 | 56.7 | 71.4 |
| Video-STR | 32 | 39.3 | 37.8 | 46.5 | 37.9 | 58.1 | 72.1 |

## 5.3 ABLATION STUDY

**Effectiveness of Different Components.** To demonstrate the contributions of each component of our proposed framework, we incrementally incorporate individual components of the framework and obtain three variants of the model: 1) w/o-graph-based reasoning mechanism, which eliminates the graph-based reasoning mechanism in the training and inference stage; 2) w/o-spatial subset, which removes the spatial understanding subset of the STV-205k in the training process; 3) w/o-temporal subset, which excludes the temporal understanding subset of the STV-205k in the training process. The evaluation results are illustrated in Table. 2, confirming the effectiveness of the graph-based reasoning mechanism. Secondly, removing spatial data leads to a notable degradation in spatial reasoning, while w/o-temporal data results in a clear decline in temporal understanding. These findings emphasize the particular effectiveness of our constructed dataset.

Table 3: Ablation study on different components.

| Method | Frames | Spatio-Temporal | | Spatial Reasoning | | Temporal Reasoning | |
|---|---|---|---|---|---|---|---|
| | | STI-Bench | V-STaR | VSI-Bench | SPAR-Bench | Video-MME | Temp-Compass |
| w/o-$R_{format}$ | 32 | 38.4 | 37.5 | 46.0 | 37.6 | 57.3 | 71.8 |
| w/o-$R_n$ | 32 | 37.7 | 36.5 | 45.6 | 37.6 | 56.9 | 71.5 |
| w/o-$R_e$ | 32 | 37.3 | 36.7 | 45.7 | 37.0 | 57.2 | 71.5 |
| w/o-$R_l$ | 32 | 38.8 | 37.4 | 45.8 | 37.9 | 57.6 | 72.0 |

**Effectiveness of Different Rewards.** We further validate the effectiveness of different reward functions. The results are shown in Table. 3. It can be observed that $R_e$ has the greatest impact on model performance, as it captures the relationships between objects and guides the model to understand multi-object topological layouts. $R_{format}$ provides a modest improvement by enforcing the output format, thereby preventing invalid or malformed responses. $R_l$ constrains the output length, which helps avoid both overthinking and underthinking, and thus effectively improves the performance.

Table 4: Accuracy of numerical predictions.

| Model | STI | VSI |
|---|---|---|
| Qwen2.5-VL-7B-Instruct | 38.1 | 33.9 |
| SpaceR (Ouyang et al., 2025) | 36.5 | 47.5 |
| Video-STR | 40.2 | 48.4 |

## 5.4 QUANTITATIVE ANALYSIS

**Accuracy of numerical QA pairs.** Considering that most of the benchmarks consist of multiple-choice questions, the model can achieve a certain level of accuracy even through random guessing, making it difficult to determine whether the model relies on memorization or true understanding. To more accurately assess the model's spatial understanding, we converted some of the multiple-choice QA in the benchmarks into numerical QA and evaluated the accuracy. As shown in the Table 4, our model demonstrates a substantial improvement in accuracy, indicating a genuine enhancement in spatio-temporal understanding.

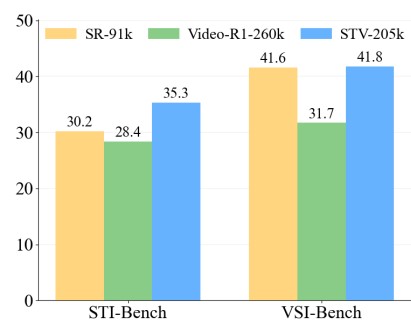

Figure 5: SFT using different datasets.

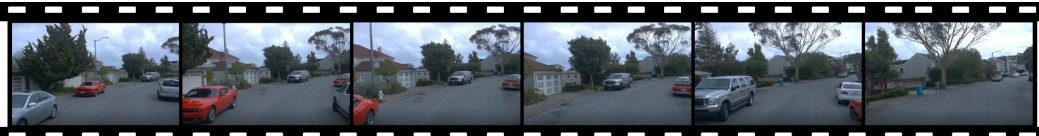

Q: What is the positional relationship of the red car relative to the white van on the left side of the road ?

A. Left  B. Right  C. Front  D. Back  E. Up

<think>Let me think through this step by step. … This is a driving perspective on a road with two lanes, where the left side is the oncoming lane. In the first frame, we see four vehicles, including a red car in the left lane and three other white vehicles. In the second frame, the white car on the left disappears from view. In the third frame, the red car disappears from view, and other vehicles appear on the right side. In the fourth frame, we can see that it is a van, and subsequently no other vehicles appear in the left lane. Considering the distribution of multiple vehicles in the visual sequence, it can be determined that the red car is in front of the white van… Let's summarize the spatial graphs of the multiple objects for the frames:
<graph>{"car":[[-2.0, 0.5, 2.4], [-1.3, 0.4, 7.0], [1.2, 0.5, 2.5]], "van":"[[0.3, 0.7, 16.0]]"}, {"car":[[-1.2, 0.4, 4.6], [1.4, 0.5, 4.2]], "van":[[0.2, 0.7, 12.0]]}, …</graph></think><answer>C</answer>

Base model response (Qwen2.5-VL-7B-Instruct): A

(a): Comparison on object spatial relation

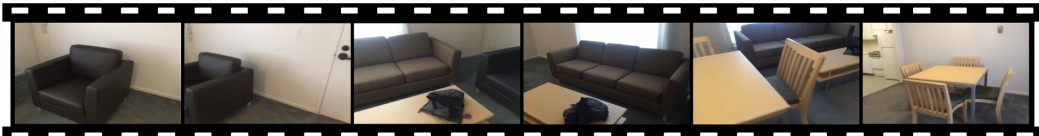

Q: What is the order of the objects' first appearance in the object list ?

A. Chair, desk, window, refrigerator, door  B. Refrigerator, door, chair, desk, tv, door

C. Sofa, door, desk, window, chair, refrigerator  D. Desk, chair, refrigerator, door, window

<think>Let me think through this step by step. … This is an indoor video. The black sofa is next to the door, and beside it there is a longer brown sofa. Behind the brown sofa, there is a window, and in front of it, there is a coffee table. Nearby, there are four chairs surrounding another tall table, and the refrigerator is in the kitchen.… we can estimate the object distribution by the spatial graph sequence: <graph>{"sofa":[[-0.2, 0.3, 1.1]]}, {"sofa":[[-0.4, 0.3, 1.2]], "door":[[0.1, 0.6, 0.9]]}, …</graph></think><answer>C</answer>

Base model (Qwen2.5-VL-7B-Instruct): B

(b): Comparison on appearance order

Figure 6: Two samples from the STI-Bench. (a) Comparison on object spatial relation; (b) Comparison on appearance order.

**Comparison with Other Datasets.** To further evaluate the dataset we designed, we compared the performance of the SFT using different datasets. Specifically, we respectively sample training data from Video-R1-260k (Feng et al., 2025), SR-91k (Ouyang et al., 2025) and STV-205k using uniform sampling. The results are shown in Fig. 5.

## 5.5 QUALITATIVE ANALYSIS

We present two cases from STI-Bench in Fig. 6 to show the process of the thinking of our model in spatio-temporal reasoning. From the Fig. 6 (a), we can see that our model demonstrates a clear qualitative superiority over the base model on topology reasoning, explicit spatial distribution, and accurate results. Our model leverages imagined graphs to understand the surrounding environment, explicitly reasoning about the distribution of multiple objects and inferring the spatial relationship between "red car" and "white van". In contrast, the base model Qwen2.5-VL-7B-Instruct directly outputs an answer, but due to the lack of spatial reasoning, it produces an incorrect result. We additionally present the results of another task in Fig. 6 (b). By reasoning across multiple frames of the video, our model precisely infers object motion trends through the spatio-temporal reasoning mechanism in the thinking process. By contrast, the base model misjudges the motion information of objects, leading to an erroneous prediction.

## 6 CONCLUSION

In this work, we present Video-STR, a novel reinforcement learning framework specifically designed to advance video spatio-temporal reasoning in MLLMs. Video-STR integrates graph reasoning directly into the model's thinking process, while employing GRPO with verifiable rewards to train the model. This mechanism enables the model to effectively capture complex multi-object distributions and layouts, which are often neglected by existing approaches. To facilitate both training and comprehensive evaluation, we introduce the STV-205k dataset, a large-scale dataset of diverse spatio-temporal QA pairs covering different scenarios. Extensive experiments demonstrate that Video-STR consistently outperforms previous spatio-enhanced and time-series modeling approaches, achieving a substantial improvement over GPT-4o on STI-Bench. These results highlight the effectiveness of integrating graph reasoning with RLVR for comprehensive video spatio-temporal reasoning. Looking forward, we aim to extend Video-STR to more complex, real-world scenarios, explore its applicability across richer modalities, and expand the dataset to encompass increasingly diverse tasks, paving the way for broader and deeper spatio-temporal reasoning in MLLMs.

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

# A    PROOF OF PROPOSED THEOREM

The center positions of the $n$ objects is represented as $\boldsymbol{x}_1, \cdots, \boldsymbol{x}_n$, and their categories are $c_1, \cdots c_n$. Then we can define a weighted complete graph (or any chosen edge subset) by

- Vertex Set: $V = \{v_1, \cdots, v_n\}$
- Label: $l(i) = c_i$
- Edge value: $d_{ij} = \|x_i - x_j\|$

**Statement for Theorem 1:** Denote this structure as $\Phi(X, C)$, where $X = (x_1, \cdots, x_n)$ and $C = (c_1, \cdots, c_n)$. Then for the rotation matrix $\mathcal{R} \in SO(3)$ (i.e., $\mathcal{R}^\top \mathcal{R} = I$ and $\det \mathcal{R} = 1$),

$$\Phi(\mathcal{R}X, C) = \Phi(X, C),$$

i.e., the graph (its labels and all edge weights) is invariant under rotations.

*Proof.* For any $i, j$, we have

$$\|\mathcal{R}x_i - \mathcal{R}x_j\| = \|\mathcal{R}(x_i - x_j)\|.$$

Since $\mathcal{R}^\top \mathcal{R} = I$, it follows that

$$\|\mathcal{R}(x_i - x_j)\| = (x_i - x_j)^\top \mathcal{R}^\top \mathcal{R}(x_i - x_j) = \|x_i - x_j\|.$$

Taking the square root yields $\|\mathcal{R}x_i - \mathcal{R}x_j\| = \|x_i - x_j\|$. Hence, all edge weights are preserved under rotations. Hence each edge weight is preserved under rotation: $w'_{ij} = w_{ij}$. Moreover, the vertex labels $\ell(i) = c_i$ are unaffected by rotation (the category of an object does not change). Therefore $\Phi(\mathcal{R}X, C)$ and $\Phi(X, C)$ have identical vertices, labels, edge sets, and edge weights, and thus

$$\Phi(\mathcal{R}X, C) = \Phi(X, C).$$

Let $\Phi = R_e$, The equation still holds.

$\square$

**Statement:** Let a center point be represented by its absolute coordinate $x_i \in \mathbb{R}^2$. For any nontrivial rotation $\mathcal{R} \in SO(3)$ with angle $\theta \neq 0 \pmod{2\pi}$, the absolute-coordinate representation is not rotation-invariant in general.

*Proof.* For a nontrivial planar rotation $\mathcal{R}$, the only fixed point is the origin. Indeed, suppose $\mathcal{R}x_i = x_i$. Then

$$(\mathcal{R} - I)x_i = \mathbf{0}.$$

Since $\mathcal{R} - I$ is invertible when $\theta \neq 0$, it follows that $x_i = \mathbf{0}$. Therefore, for any $x_i \neq \mathbf{0}$, we have $\mathcal{R}x_i \neq x_i$. Let $f$ be any feature that depends on the absolute coordinate (for example, $f(x_i) = x_i$).

Thus, for any $x_i \neq \mathbf{0}$, we have $\mathcal{R}x_i \neq x_i$. Moreover,

$$f(x_i) = x_i, \quad f(\mathcal{R}x_i) = \mathcal{R}x_i.$$

Combining these, it follows that

$$f(\mathcal{R}x_i) \neq f(x_i).$$

Hence, $f$ is not invariant under rotations.

$\square$

**Statement for Theorem 2.** The proposed graph similarity reward function can reflect the model's level of scene understanding.

**Proof.** By directly substituting the results into the reward function, we find that the reward attains a value of 1 when the nodes are in complete agreement.

# B  BENCHMARKS

- STI-Bench (Li et al., 2025c) investigates spatial-temporal based on real-world video data recorded in desktop, indoor, and outdoor environments. It proposes eight distinct tasks with over 2,000 QA pairs grouped into eight categories (including dimensional measurement, displacement and path length, ego-centric orientation, spatial relations, speed and acceleration, and trajectory description).

- V-STaR (Cheng et al., 2025) is a benchmark specifically designed to comprehensively assess the spatio-temporal reasoning capabilities of Video-LLMs. Constructed from 2094 videos covering 9 diverse domains—including Entertainment, Daily Life, Sports, Tutorials, and more—it provides coarse-to-fine Chain-of-Thought question chains and over 16,793 object bounding box annotations.

- VSI-Bench (Yang et al., 2025a) offers a large-scale resource for examining the visual–spatial reasoning skills of MLLMs. Built from 288 authentic indoor videos across a variety of locations—ranging from households to offices and factories—it contributes more than 5,000 question–answer pairs that directly test a model's capacity for spatial reasoning.

- SPAR (Zhang et al., 2025b) is a benchmark for multi-view spatial understanding. Encompassing more than 7,000 QA pairs, it spans tasks that range from simple perceptual recognition to highly complex spatial reasoning. The dataset is structured into single-view and multi-view scenarios, thereby enabling evaluation under different spatial perspectives.

- Video-MME (Fu et al., 2025) is a comprehensive benchmark for general video understanding rather than focusing exclusively on spatial aspects. It features 900 video samples and 2,700 carefully designed multiple-choice questions (three per video), spanning a broad selection of domains and activities. To ensure fairness, textual subtitles are not included in the evaluation process.

- TempCompass (Liu et al., 2024b) mainly focus on temporal reasoning. Comprising 410 videos and 7,540 questions, it is intended to measure how well MLLMs can track, interpret, and reason about temporal dynamics in the videos.

# C  BASELINES

- GPT-4o (Hurst et al., 2024): GPT-4o is a multimodal model that processes text, audio, image, and video with near real-time responses. It matches GPT-4 Turbo in English, performs better in non-English, and provides stronger vision and audio while being faster and cheaper.

- Gemini-2.0-Flash (Team et al., 2024): Gemini 2.0 Flash is a lightweight, compute-efficient multimodal model built for speed and scalability with minimal quality loss. It handles millions of tokens across text, audio, and video, achieving near-perfect long-context retrieval, strong performance in long-document and long-video QA, and efficiency for real-world applications.

- Claude-3.7-Sonnet (Anthropic, 2025): Claude 3.7 Sonnet is Anthropic's hybrid reasoning model that combines fast response with extended step-by-step thinking. It is optimized for coding, reasoning, and agentic workflows, supports long-context inputs of up to 200,000 tokens, and is available through Anthropic's API, Amazon Bedrock, and Google Cloud Vertex AI.

- Gemini-2.5-pro (Comanici et al., 2025):Gemini 2.5 Pro is the most advanced model in the Gemini 2.X family, delivering state-of-the-art performance on coding and reasoning benchmarks. It integrates long-context processing, multimodal understanding, and extended reasoning, with the ability to handle up to three hours of video content. This combination enables complex agentic workflows that require both deep reasoning and multimodal input.

- Qwen2.5-VL-7B-Instruct (Bai et al., 2025):Qwen2.5-VL-7B-Instruct is a lightweight model optimized for efficiency in resource-constrained environments. It retains key multimodal abilities such as document parsing, chart understanding, and video analysis, providing strong vision-language reasoning at lower cost.

| | |
|---|---|
| Object Counting | What is the number of {object}(s) in this video? |
| Relative Direction | Standing beside the {object_1}, should the projector screen be described as being on my left, right, or at my back? (An object is regarded as being at my back if facing it requires a turn of at least 135 degrees.) |
| Relative Distance | From the center points of the {object_1} and the {object_2}, what is the direct distance separating them in meters? |
| Appearance Order | How do the categories {object_list} in sequence within the video? |
| Object Size | In meters, what is the maximum dimension (length, width, or height) of the {object}? |
| Motion Tracking | At the {timestamp_1}, the {object}'s boundingbox is [118, 97, 393, 319] in a 640x480 resolution video, what's the boundingbox of the {object} at the {timestamp_2}? |
| Object Localization | This is a single randomly chosen frame, estimate the 3D bounding box of the {object} in camera coordinates. |
| Displacement | Between {timestamp_1} and {timestamp_2}, what is the most probable straight-line displacement of the {object} across two frames? |

Figure 7: Question templates of STR-205k.

- MiniCPM-V-2.6 (Yao et al., 2024):MiniCPM-V 2.6 is an efficient multimodal model designed for end-device deployment. It delivers strong OCR, high-resolution image perception, multilingual support, and low hallucination, while achieving performance comparable to larger cloud-based models at much lower cost.

- VideoLLaMA3-7B (Zhang et al., 2025a): VideoLLaMA3-7B is a vision-centric multimodal model for image and video understanding. Trained primarily on large-scale image-text data and refined through multi-stage alignment and fine-tuning, it supports variable-resolution inputs and compact video representations, achieving strong performance across image and video benchmarks.

- Video-R1 (Feng et al., 2025):Video-R1 is a multimodal model designed for video reasoning, extending the R1 reinforcement learning paradigm with temporal modeling and curated image–video datasets. Using the T-GRPO algorithm and large-scale reasoning data, it achieves notable gains on benchmarks such as VideoMMMU, MVBench, TempCompass, and surpasses GPT-4o on VSI-Bench.

- VideoChat-R1 (Li et al., 2025b):VideoChat-R1 is a video multimodal model enhanced through reinforcement fine-tuning with GRPO, targeting spatio-temporal perception while preserving general chat ability. It achieves state-of-the-art results on temporal grounding and object tracking, and shows consistent improvements on video QA benchmarks such as VideoMME, MVBench, and Perception Test.

- SpaceR (Ouyang et al., 2025):VideoChat-R1 adopts the SpaceR framework to enhance video spatial reasoning through Spatially-Guided RLVR and a new 151k dataset. It achieves state-of-the-art results on spatial reasoning benchmarks such as VSI-Bench, STI-Bench, and SPAR-Bench, surpassing GPT-4o and matching Gemini-2.0-Flash, while maintaining strong general video understanding.

# D DATASET CONSTRUCTION

**QA Generation.** As illustrated in Fig. 7, we incorporate the corresponding objects, timestamps into the question templates.

## E  ADDITIONAL IMPLEMENTATION DETAILS

**Data Sampling.** We perform uniform sampling of QAs across various task types, resulting in 12,000 QA pairs. Furthermore, to prevent the model from degrading in general video understanding during training, we incorporate a subset with 3k QA of data from Video-R1-260k (Feng et al., 2025) to preserve its generalization capability in video understanding.

## F  ADDITIONAL DISCUSSION.

**Why not use SFT?** Instead of using SFT, we adopt RLVR because RLVR directly optimizes for task-specific objectives and desired behaviors, allowing the model to learn from feedback signals rather than just mimicking labeled data. This approach better aligns the model with complex reward functions, improves its adaptability, and enhances performance on tasks where simple supervised learning may not capture nuanced preferences or long-term outcomes (Ouyang et al., 2025; Feng et al., 2025).

We further conducted evaluations on LongVideoBench (Wu et al., 2024), which serves as a fundamental benchmark for video understanding, and the results are shown in the Table. 5. It can be observed that our model does not introduce any performance degradation and retains its original general video understanding capability.

Table 5: Results on general video understanding task.

|  | LongVideoBench |
| --- | --- |
| Qwen2.5-VL-7B-Instruct (Base) | 54.0 |
| Ours (SFT) | 53.5 |
| Ours | 54.3 |

## G

Comparison with Other Methods. The model's performance indeed depends on both the training data and the algorithm. It is worth noting that algorithms and datasets are highly coupled. For example, we used graph reasoning, so we collected graph data. Similarly, SpaceR leverages a grid map for reasoning, so it collects grid map data. We further train SpaceR using STV-205k and evaluate the performance, but exclude the map imagination mechanism. The results are shown in the table below. It can be seen that the performance is worse than ours, but improved compared to the model trained using SR-91k in Table. 1, indicating that both our algorithm and data are effective.

## H  PROMPT TEMPLATE

Fig. 8 shows the prompt template for training and inference. Fig. 9 illustrates the prompt template for the graph reasoning mechanism.

## I  TRAINING CURVES

Fig. 10 shows the training curves of our proposed framework. It is evident that the accuracy reward shows a generally increasing trend, indicating that the model steadily improves its ability to generate correct answers under reinforcement learning. The performance changes of our model on STI-Bench are also illustrated. An upward trend can be observed, which validates the effectiveness of the verifiable reward we designed.

Table 6: Results of baseline using STV-205k.

| Method | Dataset | STI | V-STaR | VSI | SPAR | VMME | TC |
|--------|---------|-----|--------|-----|------|------|-----|
| Qwen2.5-VL-7B-Instruct | - | 34.7 | 35.2 | 33.0 | 33.1 | 56.3 | 71.1 |
| SpaceR | STV-205k | 37.2 | 35.3 | 45.9 | 37.6 | 57.5 | 71.3 |

---

**Prompt Template for Training and Inference**

{QUESTION_TEMPLATE}
Please think about this question as if you were a human pondering deeply. Engage in an internal dialogue using expressions such as 'let me think', 'wait', 'Hmm', 'oh, I see', 'let's break it down', etc, or other natural language thought expressions. It's encouraged to include self-reflection or verification in the reasoning process. Provide your detailed reasoning between the <think> </think> tags, and then give your final answer between the <answer> </answer> tags.

{TYPE_TEMPLATE}
"multiple choice": " Please provide only the single option letter (e.g., A, B, C, D, etc.) within the <answer> </answer> tags.",
"numerical": " Please provide the numerical value (e.g., 42 or 3.1) within the <answer> </answer> tags.",
"OCR": " Please transcribe text from the image/video clearly and provide your text answer within the <answer> </answer> tags.",
"free-form": " Please provide your text answer within the <answer> </answer> tags.",
"regression": " Please provide the numerical value (e.g., 42 or 3.14) within the <answer> </answer> tags."
"IoU": " Please provide the bounding box (e.g., [120, 80, 180, 170]) within the <answer> </answer> tags."

Figure 8: Prompt Template for Training and Inference.

---

**Prompt Template for Graph Reasoning**

{GRAPH_TEMPLATE}
It's encouraged to include self-reflection or verification in the reasoning process. If generating a series of locations of the objects in the image or video can help you answer the question, you could follow the below steps to generate the coordinates of the objects' center points in <graph> </graph> tags. [Steps] Identify specific objects within the visual **scene**, understand the spatial arrangement of the scene, and estimate the center point of each object, assuming the ego camera as the reference frame, i.e., the origin of the coordinate system. These information should be summarized in <graph> </graph> tags. [Rule]1. We provide the categories to care about in this scene: {object_list}. Focus ONLY on these categories for the entire scene. Estimate the center location of each instance within the provided categories, assuming the entire scene is represented by a three-dimensional coordinate system, considering the information from all frames. The X-axis represents the left and right of the camera, with right as the positive direction. The Y-axis represents up and down, with up as the positive direction. The Z-axis represents front and back, with front as the positive direction. If a category contains multiple instances, include only the most evident and nearest one. Present the map using dict format. Here is an example: <graph>{graph_example}</graph>. If you generate the locations of objects, please put it in <graph> </graph> tags. Provide your detailed reasoning process between the <think> </think> tags, and then give your final answer between the <answer> </answer> tags.

Figure 9: Prompt Template for Graph Reasoning.

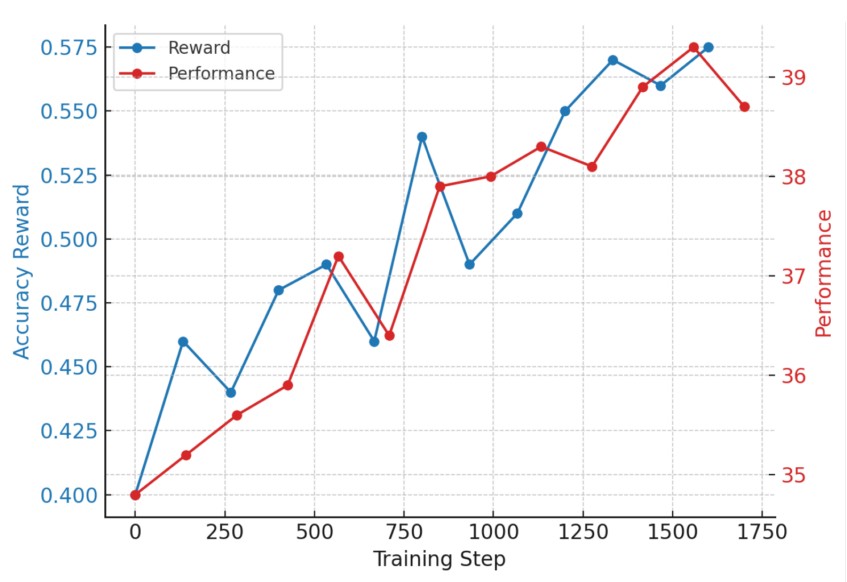

Figure 10: Training curves.

