# OpenReview forum: "Video-STR: Reinforcing MLLMs in Video Spatio-Temporal Reasoning with Relation Graph"
_ICLR.cc/2026/Conference — Submitted to ICLR 2026_

### Official Review · Reviewer_q9jw · 2025-10-30

**Soundness:** 3
**Presentation:** 3
**Contribution:** 2
**Rating:** 4
**Confidence:** 3

**Summary:**

The paper describes a dataset consisting of Question-Answer pairs about object-based tasks (such as counting, discances, movements, etc.) that were automatically generated from existing video dataset (TAO, ScanNet and KITTI). It also describes a reward design for RL-based training, based on the positions of objects as well as their positions relative to one another. It then shows that together, these two ingredients make it possible to fine-tune Qwen2.5-VL-7B to achieve relatively high recognition performance on several video understanding benchmarks (STI-bench, V-STaR, etc.). Ablations furthermore show that all the ingredients contribute positively to performance improvements on these benchmarks.

**Strengths:**

The paper is quite easy to follow. It describes an (automatically generated) dataset, a training method and a model (based on fine-tuning Qwen2.5-VL-7B), all of which will be publicly released. The model yields fairly good performance on some video understanding benchmarks, specifically those that test for an understanding of object positions and their relations.

**Weaknesses:**

I have the concern that the object-centric data and reward designs are benefitting specifically tasks that measure the ability to understand object positions and relations, without enhancing the model’s visual understanding overall. The results suggest that this is indeed the case, as model performance on tasks like STI-bench and V-STaR is comparably high, while this is not nearly as much the case for tasks like Video-MME (where the model is in fact significantly behind the state of the art).

Overall, this paper describes an engineering effort, based on automatically generated data and RL-training with object-centric rewards, to mildly improve performance on video-understanding benchmarks (and most significantly on object-centric benchmarks). I am not quite sure how this advances our understanding of visual capabilities, models and model limitations overall. I am also missing a view of the limitations of the object-centric design. For example, are there any tasks where performance degrades rather than benefits from this design?

**Questions:**

After fine-tuning the model on graph-based spatio-temporal reasoning, how is it applied to the downstream benchmark tasks? Is there additional fine-tuning involved? Any task-specific (few-shot?) prompting?

The work is reminiscent of older works on graph-based grounding, such as “Visual Genome: Connecting Language and Vision Using Crowdsourced Dense Image Annotations”, Krishna et al 2016, and many related and follow-up papers). It would be good to see a discussion on how it is related to this line of research.

(small comment:) The frames in the figures (for example, Figure 1, Figure 2) are impossible to see properly or understand in a print-out version of the paper.

---

> ### Author Response · Authors · 2025-11-24
> **Response to Reviewer q9jw**
>
> Thanks for your insightful feedback. We've consolidated your comments into 5 questions and provided responses to each.
>
> **Q1: I have the concern that the object-centric data and reward designs are benefitting specifically tasks that measure the ability to understand object positions and relations, without enhancing the model’s visual understanding overall. The results suggest that this is indeed the case, as model performance on tasks like STI-bench and V-STaR is comparably high, while this is not nearly as much the case for tasks like Video-MME (where the model is in fact significantly behind the state of the art).**
>
> **R1:** The experimental results are as expected, since our primary focus is on spatiotemporal reasoning tasks.
> For the Video-MME scenarios, we did not apply any specialized reinforcement, and therefore no significant improvement is observed.
>
> **Q2: Overall, this paper describes an engineering effort, based on automatically generated data and RL-training with object-centric rewards, to mildly improve performance on video-understanding benchmarks (and most significantly on object-centric benchmarks). I am not quite sure how this advances our understanding of visual capabilities, models and model limitations overall. I am also missing a view of the limitations of the object-centric design. For example, are there any tasks where performance degrades rather than benefits from this design?**
>
> **R2:** Your concern is reasonable, as post-training can easily cause the large model to overfit on specific tasks.
> This issue is more severe for SFT than for RL, because SFT forces the model to fit the training data, which can degrade its original capabilities, whereas RL causes less parameter distribution shift.
>
> In our experiments, we selected Video-MME and TempCompass, which are standard video reasoning benchmarks. The results on these benchmarks demonstrate that our trained model can enhance spatiotemporal understanding without compromising its fundamental reasoning capabilities.
> In contrast, models trained with SFT, while exhibiting improved spatiotemporal reasoning, show reduced general video understanding compared to the base model, indicating that their original capabilities have been compromised.
>
> We further conducted evaluations on LongVideoBench [1], which serves as a fundamental benchmark for video understanding, and the results are shown in the table below. It can be observed that our model does not introduce any performance degradation and retains its original general video understanding capability.
>
> | Model     | LongVideoBench [1] |
> |----------|-----|
> | Qwen2.5-VL-7B-Instruct (Base)     | 54.0        |
> | Ours (SFT)        | 53.5          |
> | Ours         | 54.3      |
>
> [1] Haoning Wu, et al. Longvideobench: A benchmark for long-context interleaved video-language understanding. Advances in Neural Information Processing Systems 2024.
>
> **Q3: After fine-tuning the model on graph-based spatio-temporal reasoning, how is it applied to the downstream benchmark tasks? Is there additional fine-tuning involved? Any task-specific (few-shot?) prompting?**
>
> **R3:**
> Yes, different downstream tasks require different task-specific prompts. In our experiments, the prompts depend on the benchmark and task types, and all baselines use the same prompts. No additional fine-tuning is involved in the tests.
>
> For example, the prompts for multi-choice tasks used in STI-Bench are shown below:
> Please provide only the single option letter (e.g., A, B, C, D, etc.) within the **\<answer> ... \</answer>** tags.
>
>
> **Q4: The work is reminiscent of older works on graph-based grounding, such as “Visual Genome: Connecting Language and Vision Using Crowdsourced Dense Image Annotations”, Krishna et al 2016, and many related and follow-up papers). It would be good to see a discussion on how it is related to this line of research.**
>
> **R4:**
> We appreciate the reviewer for pointing us to this high-quality related work. Visual Genome [2] leverages scene graphs as a representational framework for image understanding, enabling the modeling of semantic relationships among objects, for instance interactions such as “person riding horse”, thereby facilitating the alignment between natural language and visual content. However, it does not encompass precise spatiotemporal perception, including spatial configurations or temporal dynamics, which represents a fundamental divergence from our work.
>
> [2] Krishna, Ranjay, et al. Visual genome: Connecting language and vision using crowdsourced dense image annotations. International journal of computer vision 2017.
>
> **Q5: (small comment:) The frames in the figures (for example, Figure 1, Figure 2) are impossible to see properly or understand in a print-out version of the paper.**
>
> **R5:**
> Thank you for the reminder. We overlooked the print-out scenario, and we will provide a clearer version of the paper with high-resolution frames.

---

> > ### Comment · Reviewer_q9jw · 2025-11-27
> >
> > Thank you for the clarifications.
> >
> > A high-level concern that I feel remains is that the main take-home message of the paper is: Fine-tuning on automatically generated surrogate information, such as precise localizations, improves performance on benchmark tasks that test for precisely this kind of information. While it is good that this can be done without degrading performance on other tasks, it is not clear to me how substantial or surprising the finding really is, and how much real-world down-stream tasks will really need to rely on this kind of information.

---

> > > ### Author Response · Authors · 2025-12-04
> > > **Response to the follow-up question**
> > >
> > > Thank you for your feedback. MLLMs are increasingly employed as end-to-end solutions for Embodied AI [1-4] and Autonomous Driving [5-8]. Such tasks require MLLMs to understand 3D space and time, going beyond conventional 2D visual perception. For robotic and vehicular systems to predict optimal action strategies, precise spatio-temporal world understanding is essential.
> > >
> > > [1] Zhili Cheng, et al. Embodiedeval: Evaluate multimodal llms as embodied agents. arXiv preprint arXiv:2501.11858 (2025).
> > >
> > > [2] Chengshu Li, et al. Behavior-1k: A human-centered, embodied ai benchmark with 1,000 everyday activities and realistic simulation. arXiv preprint arXiv:2403.09227 (2024).
> > >
> > > [3] Manling Li, et al. Embodied agent interface: Benchmarking llms for embodied decision making. Advances in Neural Information Processing Systems 37 (2024): 100428-100534.
> > >
> > > [4] Oier Mess, et al. Calvin: A benchmark for language-conditioned policy learning for long-horizon robot manipulation tasks. IEEE Robotics and Automation Letters 7.3 (2022): 7327-7334.
> > >
> > > [5] Jyh-Jing Hwang, et al. Emma: End-to-end multimodal model for autonomous driving. arXiv preprint arXiv:2410.23262 (2024).
> > >
> > > [6] Yi Xu, et al. Vlm-ad: End-to-end autonomous driving through vision-language model supervision. arXiv preprint arXiv:2412.14446 (2024).
> > >
> > > [7] Zhenhua Xu, et al. Drivegpt4: Interpretable end-to-end autonomous driving via large language model. IEEE Robotics and Automation Letters (2024).
> > >
> > > [8] Jianhao Yuan, et al. Rag-driver: Generalisable driving explanations with retrieval-augmented in-context learning in multi-modal large language model. arXiv preprint arXiv:2402.10828 (2024).

---

### Official Review · Reviewer_hMWm · 2025-10-31

**Soundness:** 2
**Presentation:** 2
**Contribution:** 2
**Rating:** 4
**Confidence:** 3

**Summary:**

This paper presents a graph-based reinforcement learning method for Video Spatio-Temporal Reasoning which models inter-object relationships using relation graphs. It introduces a graph-based reasoning mechanism in the Group Relative Policy Optimization (GRPO) [Shao et al. 2024] for inferring the underlying spatio-temporal topology of scenarios during the thinking process. The paper proposes STV-205k dataset with 205k question-answering pairs for model training and reports SOTA performance on various benchmarks including STIBench.

**Strengths:**

The motivation and problem definition are valid.

The graph based reasoning approach is interesting and seems novel, although I am not aware of all the research in the area of video reasoning. Nevertheless, a graph-based approach appears to be a viable direction for video reasoning.

The idea of generating a QA dataset (STV-205K) from existing datasets is innovative and useful.

Reported experimental results are promising.

**Weaknesses:**

My biggest concerns include limited contribution, lack of clarity and the possible unfair comparison in experiments. Details below.

Graph-based spatio-temporal reasoning is not a new approach and has been done previously. This work appears to be a combination of existing techniques with limited methodological novelty.

The very first sentence of the abstract is grammatically incorrect. Overall, the writing and presentation of the paper can certainly be improved as I find it quite confusing. Which “model” is being referred to at each place? And how distances are calculated in images? How do you get Euclidean distances from images/videos? Are they in pixels and if so, how meaningful are pixel based distances in a graph?

What is X in Eqn.(8)? It should be defined after the equation and not only in the Appendix. Going to the proof in the Appendix, X seems to be the object locations and not distances and locations should not be rotation invariant. On the other hand, graph edge weights being Euclidean distances will remain rotation invariant. Also, images/video frames are 2D images, why is the rotation R \in SO(3)?

Also, I did not notice the proof of Theorem 2 in the Appendix (maybe I overlooked but there were no numbers with Theorem proof in the Appendix). I am also not sure if these two theorems are worth mentioning at all.

At this point, I should also highlight the ambiguity in the proposed dataset description. It is not explicitly mentioned which data modality and labels are obtained from the three datasets (TAO, ScanNet and KITTI) e.g. ScanNet and KITTI also have depth images/point clouds. There is no mention of 3D, depth or point cloud in the paper so I assume the paper is about conventional video reasoning but then “Euclidean distance” is mentioned at many places which is confusing. Yes, the Euclidean distances can be in pixel units but that would be meaningless in a scene graph e.g. the chair is two pixels to the right of the table.

Referring to “The model is prompted to generate a graph” on page 5, which model is prompted? The following sentence is also confusing where “model” is used twice. Are they both the same model or two different models?

On page 2 it is mentioned that “we extend the Group Relative Policy Optimization (GPRO)” but in Section 4.3, it is not clearly mentioned how GPRO is extended in this paper.

The results in Table 1 look very good but I am wondering if the improvement is mainly due to the additional training data from newly proposed STV-205k dataset. Since the other methods did not use this additional dataset, the comparison is unfair.

Page 7: “SFT achieves localized improvements”. What does “localized improvement” mean?

In Fig.4(b), the improvement on “Static” sub-task is quite substantial compared to “Temporal”. This makes me wonder how the graph encodes temporal information.

In Sec. 5.3, “Video-260k”, “SR-91k” are mentioned, but no reference or any further information is given about these datasets.

Overall, the proposed method may have merit but that is obscured by unclear presentation. Moreover, unfair comparison with existing works also undermines its credability.

**Questions:**

Please see weaknesses above.

---

> ### Author Response · Authors · 2025-11-24
> **Response to Reviewer hMWm (part 1/2)**
>
> We thank you for your time and effort in reviewing our paper. We’ve consolidated your comments into 11 questions and provided responses to each.
>
> **Q1: Graph-based spatio-temporal reasoning is not a new approach and has been done previously. This work appears to be a combination of existing techniques with limited methodological novelty.**
>
> **R1:** Our novelty lies in the precise spatiotemporal reasoning of MLLMs, a domain that has not been fully explored. Although graphs are not a new concept, we are the first to enable large models to “imagine” graphs during post-training, using this to supervise their thinking process and achieve a deeper understanding of the physical world.
>
> **Q2: The very first sentence of the abstract is grammatically incorrect. Overall, the writing and presentation of the paper can certainly be improved as I find it quite confusing. Which “model” is being referred to at each place? And how distances are calculated in images? How do you get Euclidean distances from images/videos? Are they in pixels and if so, how meaningful are pixel based distances in a graph?**
>
> **R2:** Thank you for pointing out the error. We have corrected it and will polish the entire manuscript. In this paper, all references to ‘model’ refer to multimodal large language models (MLLMs).
>
> The distance mentioned in the paper is the Euclidean distance in the real physical world, not the pixel distance.
> In our constructed STV-205k dataset, we obtain Euclidean distance information from the annotations of the two original 3D perception datasets: KITTI and Scannet.
>
> Only the QA pairs derived from TAO, which is a 2D object tracking dataset, involve image pixels. These include the motion tracking and object localization tasks, and the ground-truth values are obtained from the TAO annotations.
>
> During training, the designed graph reasoning is applied only to the QA data derived from Scannet and KITTI, and not to TAO.
>
> **Q3:** What is X in Eqn.(8)? It should be defined after the equation and not only in the Appendix. Going to the proof in the Appendix, X seems to be the object locations and not distances and locations should not be rotation invariant. On the other hand, graph edge weights being Euclidean distances will remain rotation invariant. Also, images/video frames are 2D images, why is the rotation R \in SO(3)?
>
> **R3:** Your understanding is correct; X represents the 3D coordinates of the objects' positions.
> What we proved in the appendix is that the edges are rotation-invariant, which corresponds to $R_e$ in the reward functions.
> Similar to R2, the ‘distance’ here refers to the distance in the physical world, and the coordinates of the objects are relative to the ego camera, i.e., the coordinate system is centered at the ego camera. Therefore, changes in the camera viewpoint will result in a rotation of the coordinate system.
>
> **Q4: Also, I did not notice the proof of Theorem 2 in the Appendix (maybe I overlooked but there were no numbers with Theorem proof in the Appendix). I am also not sure if these two theorems are worth mentioning at all.**
>
> **R4:** This point is indeed not very crucial; we provided it in the appendix, but it was not very noticeable. We have marked it clearly in the updated version.
>
> **Q5: At this point, I should also highlight the ambiguity in the proposed dataset description. It is not explicitly mentioned which data modality and labels are obtained from the three datasets (TAO, ScanNet and KITTI) e.g. ScanNet and KITTI also have depth images/point clouds. There is no mention of 3D, depth or point cloud in the paper so I assume the paper is about conventional video reasoning but then “Euclidean distance” is mentioned at many places which is confusing. Yes, the Euclidean distances can be in pixel units but that would be meaningless in a scene graph e.g. the chair is two pixels to the right of the table.**
>
> **R5:** For all three datasets, we only used the video information and did not use the point clouds. KITTI and ScanNet are 3D perception datasets and provide ground-truth 3D annotations. Consequently, spatial information, such as Euclidean distances, can be derived. The graph reasoning is conducted based on this 3D information. For the TAO dataset, which is a 2D object tracking dataset, we utilized pixel-level annotations to construct the motion tracking and object localization tasks in the dataset, where the model is required to predict 2D bounding boxes.
>
> **Q6: Referring to “The model is prompted to generate a graph” on page 5, which model is prompted? The following sentence is also confusing where “model” is used twice. Are they both the same model or two different models?**
>
> **R6:** As stated in R2, the model we refer to is the MLLM that we have trained. We did not use any other models.

---

> > ### Author Response · Authors · 2025-11-24
> > **Response to Reviewer hMWm (part 2/2)**
> >
> > **Q7: On page 2 it is mentioned that “we extend the Group Relative Policy Optimization (GPRO)” but in Section 4.3, it is not clearly mentioned how GPRO is extended in this paper.**
> >
> > **R7:** GRPO is a rule-based RL algorithm, that uses reference answers to compute advantages.
> > We introduce a graph reasoning mechanism into the thinking process, enabling the MLLM to imagine the layouts of objects and thereby understand the topological information in space. We also designed task-specific rewards to compute the advantages.
> >
> > **Q8: The results in Table 1 look very good but I am wondering if the improvement is mainly due to the additional training data from newly proposed STV-205k dataset. Since the other methods did not use this additional dataset, the comparison is unfair.**
> >
> > **R8:** We understand your concern. The model’s performance indeed depends on both the training data and the algorithm.
> > It is worth noting that **algorithms and datasets are highly coupled.**
> > For example, we used graph reasoning, so we collected graph data. Similarly, SpaceR leverages a grid map for reasoning, so it collects grid map data. We further train SpaceR using STV-205k, but exclude the map imagination mechanism. The results are shown in the table below. It can be seen that the performance is worse than ours, but improved compared to SR-91k built using SpaceR, indicating that both our algorithm and data are effective.
> >
> > | Method                        | Dataset  | STI  | V-STaR | VSI  | SPAR | VMME | TC   |
> > |------------------------------|----------|------|--------|------|------|------|------|
> > | Qwen2.5-VL-7B-Instruct       | -        | 34.7 | 35.2   | 33.0 | 33.1 | 56.3 | 71.1 |
> > | SpaceR                       | STV-205k | 37.2 | 35.3   | 45.9 | 37.6 | 57.5 | 71.3 |
> >
> >
> > **Q9: Page 7: “SFT achieves localized improvements”. What does “localized improvement” mean?**
> >
> > **R9: ** "localized improvement" means that SFT primarily enhances model performance only in the vicinity of the training examples. Because SFT serves the model to directly imitate labeled input–output pairs, the model tends to memorize specific answers rather than acquire generalizable reasoning skills. Thus, SFT improvements are local to the distribution covered by the fine-tuning data and cannot transfer robustly to unseen or out-of-distribution scenarios, making SFT prone to overfitting.
> >
> > **Q10: In Fig.4(b), the improvement on “Static” sub-task is quite substantial compared to “Temporal”. This makes me wonder how the graph encodes temporal information.**
> >
> > **R10:** The performance on static spatial understanding exceeds that on dynamic temporal reasoning primarily because temporal reasoning is inherently more challenging and additionally relies on spatial understanding as its foundation.
> > We do not explicitly encode temporal information into the graph; instead, we implicitly supervise temporal relations by computing the average reward across multiple video frames, as illustrated in Eq. 7 of the paper.
> >
> >
> > **Q11: In Sec. 5.3, “Video-260k”, “SR-91k” are mentioned, but no reference or any further information is given about these datasets.**
> >
> > **R11:**
> > Thank you for pointing this out; we have corrected it in the updated version of the paper.
> > Video-R1-260k [1] is a multi-modal reasoning question–answering dataset containing 260k QA pairs, covering general, chart, OCR, math, knowledge, and spatial tasks.
> > SR-91k [2] is a spatial reasoning dataset that spans six spatial reasoning tasks (e.g., relative direction, object size, and absolute distance).
> >
> > [1] Kaituo Feng, et al. Video-r1: Reinforcing video reasoning in mllms. Neurips 2025.
> >
> > [2] Kun Ouyang, et al. SpaceR: Reinforcing MLLMs in Video Spatial Reasoning. arXiv preprint arXiv:2504.01805, 2025.

---

> > > ### Comment · Reviewer_hMWm · 2025-11-26
> > > **Discussion on Response part 2**
> > >
> > > I appreciate the additional clarifications but as I said in my previous discussion, the clarifications are far more than one would expect in a rebuttal. It requires substantial revisions to the paper.
> > >
> > > The results in R8 look interesting and I understand that algorithms and data are tied i.e. not all algorithms are designed to use exactly the same type of data. But this makes the overall comparisons between the methods complicated at the least. I would like to hear the opinion of the other reviewers on this.
> > >
> > > R10. Not explicitly encoding temporal information into the graph when the aim is "sptio-temporal reasoning" does not seem like an optimal approach.

---

> > > > ### Author Response · Authors · 2025-12-04
> > > > **Response to the follow-up question**
> > > >
> > > > Thank you once again for your insightful feedback on our submission. We provide our responses below.
> > > >
> > > > **Q: Not explicitly encoding temporal information into the graph when the aim is "sptio-temporal reasoning" does not seem like an optimal approach.**
> > > >
> > > > Explicitly encoding temporal information into the graph would couple the spatial and temporal information, making them difficult to process simultaneously. In contrast, our approach is equivalent to allowing the model to first determine spatial locations and then predict the evolution of actions, thereby decoupling the two types of information. This has been shown to be more efficient in visual tasks [1-4].
> > > >
> > > > [1] Nuo Chen, et al. Motion and Appearance Decoupling Representation for Event Cameras. IEEE Transactions on Image Processing (2025).
> > > >
> > > > [2] Qingyu Shi, et al. Decouple and track: Benchmarking and improving video diffusion transformers for motion transfer. Proceedings of the IEEE/CVF International Conference on Computer Vision. 2025.
> > > >
> > > > [3] Shuting He, and Henghui Ding. Decoupling static and hierarchical motion perception for referring video segmentation. Proceedings of the IEEE/CVF Conference on Computer Vision and Pattern Recognition. 2024.
> > > >
> > > > [4] Rui Qian, et al. Static and dynamic concepts for self-supervised video representation learning. European conference on computer vision,  2022.

---

> > ### Comment · Reviewer_hMWm · 2025-11-26
> > **Discussion on response part 1**
> >
> > Thank you for prividing a detailed response. My questions regarding ambiguities in the method have been largely addressed and I appreciate response regarding novelty.
> >
> > Clarity: In my opinion, the paper needs more revisions than one would expect from the rebuttal phase. Reviewer 5SFE has also pointed out missing critical details.
> >
> > Novelty: I still think the contribution of the paper is limited and not to the level of a machine learning conference. Reviewer q9jw has alos pointed out that the paper is mainly an engineering effort.

---

### Official Review · Reviewer_5SFE · 2025-11-01

**Soundness:** 3
**Presentation:** 3
**Contribution:** 3
**Rating:** 4
**Confidence:** 4

**Summary:**

The paper introduces Video-STR, using graph-based reasoning and reinforcement learning to improve spatio-temporal understanding in video MLLMs. The authors build a 205k QA dataset and extend GRPO with graph rewards. Results are promising (13% gain on STI-Bench), but the work has several technical gaps and unclear design choices that weaken the contribution.

**Strengths:**

- The graph representation for multi-object scenes makes sense and has nice rotation-invariance properties
- The benchmarking across 6 different datasets shows the approach generalizes reasonably well

**Weaknesses:**

- Missing critical implementation details. For example: How do you actually detect and track objects to build these graphs?; During inference, does the model predict object locations itself, or do you use an external detector?
- 205k automatically generated QA pairs with no human verification is risky. Template-based generation could introduce systematic biases
- The paper says the model is "prompted to generate such a graph" but doesn't show how. Is it predicting coordinates in structured format?
- Which reward components actually matter? No ablation on R_format, R_n, R_e, Rl_individually
- GRPO generates 8 responses per sample - what's the actual training time and memory cost?

**Questions:**

Please answer the questions in the weaknesses section

---

> ### Author Response · Authors · 2025-11-24
> **Response to Reviewer 5SFE**
>
> We appreciate your detailed suggestions for our work. We provide the response to your concerns point by point as follows:
>
> **Q1: Missing critical implementation details. For example: How do you actually detect and track objects to build these graphs?; During inference, does the model predict object locations itself, or do you use an external detector?**
>
> **R1:**  In our method, object detection and tracking are not performed by any external detectors. Instead, all object localization and temporal association are handled internally by our trained model.
> For the construction of graphs, we achieve it by setting specific prompts. The prompts are shown in Figure 9 in the Appendix.
>
> **Q2: 205k automatically generated QA pairs with no human verification is risky. Template-based generation could introduce systematic biases**
>
> **R2:**
> When generating QA pairs, we have set up paraphrase templates to prevent systematic biases. For example, for the object counting task, we designed five question templates:
>
> 1. What's the number of the {object}s in this video?
>
> 2. How many {objects} are there in this video?
>
> 3. What is the count of {objects} shown in this video?
>
> 4. Can you tell me how many {objects} appear in this video?
>
> 5. How many {objects} does the video contain?
>
> Besides, we also applied several manual filtering steps. For multiple-choice questions, we randomized the order of the options; for numerical questions, we adjusted the value distribution by removing extreme values.
>
> **Q3: The paper says the model is "prompted to generate such a graph" but doesn't show how. Is it predicting coordinates in structured format?**
>
> **R3:** Yes, the model is architecturally guided by our designed prompts to yield coordinate information in a structured format. The prompts are presented in Figure 9 of the appendix. And the sequence of the coordinates will be wrapped into **\<graph> ... \</graph>**.
>
>
> **Q4: Which reward components actually matter? No ablation on R_format, R_n, R_e, Rl_individually**
>
> **R4:** Thanks for your suggestion. We have added an experiment to validate the effectiveness of different reward functions. The results are shown in the table below.
>
> | Method        | STI   | V-STaR | VSI   | SPAR | VM   | TC   |
> |---------------|-------|--------|-------|------|------|------|
> | w/o-$R_{format}$   | 38.4  | 37.5   | 46.0  | 37.6 | 57.3 | 71.8 |
> | w/o-$R_n$        | 37.7  | 36.5   | 45.6  | 37.6 | 56.9 | 71.5 |
> | w/o-$R_e$        | 37.3  | 36.7   | 45.7  | 37.0 | 57.2 | 71.5 |
> | w/o-$R_l$        | 38.8  | 37.4   | 45.8  | 37.9 | 57.6 | 72.0 |
>
> **Q5: GRPO generates 8 responses per sample - what's the actual training time and memory cost?**
>
> **R5:**
> The required training time was approximately 22 hours, with a mean memory cost of around 540 GB, using 8 NVIDIA H100-80GB GPUs.

---

### Official Review · Reviewer_KoMw · 2025-11-11

**Soundness:** 3
**Presentation:** 3
**Contribution:** 3
**Rating:** 8
**Confidence:** 4

**Summary:**

The authors propose Video-STR, a graph-based RL framework to model the fine-grained interactions in video, enabling advanced reasoning in scenarios with heavy physical information. To facilitate the model training, they collect a large-scale training dataset, STV-205K. The experimental results demonstrate the effectiveness of the training data, training strategy (graph-based GRPO). Video-STR outperforms existing video reasoning models on most benchmarks.

**Strengths:**

1. The paper is well-written and has a good organization.
2. The proposed Video-STR is reasonable and can truly process complicated video scenarios.
3. The STV-205K is carefully constructed and well aligned with the proposed framework.
4. The experiments are comprehensive, and the ablations are solid.

**Weaknesses:**

This paper is of good quality, and I just have a question about the setting of the dataset. It looks like there are still mostly recognition-type data in the dataset, not real reasoning data. From line 173-190. However, the design of the graph can be further explored, for example, some data like the intention of a motion.

**Questions:**

Please refer to the weakness.

---

> ### Author Response · Authors · 2025-11-24
> **Respond to Reviewer KoMw**
>
> We sincerely appreciate the reviewer's constructive feedback and positive remarks on our work. We provide the following detailed responses to your major concerns.
>
> **Q1: The design of the graph can be further explored, for example, some data like the intention of a motion.**
>
> **R1:** Thank you for your suggestion. We agree that it is worthwhile to further explore the capabilities of graphs by changing their design. Graph is a natural and powerful structure that can model relationships among multiple nodes, where each node can correspond to real-world entities such as objects, events, or actions.
> By explicitly representing these entities and the connections between them, graphs allow models to capture complex relational patterns.
>
> We use graphs to characterize the spatial relationships among multiple objects. Although the potential of graphs has not been fully leveraged, we have already observed encouraging improvements in MLLM video spatio-temporal reasoning performance.  In addition, we conduct graph-based reasoning across multiple frames in the video understanding, which implicitly facilitates the learning of motion cues.  This further enhances the model’s performance on temporal sub-tasks (e.g., displacement estimation).
>
> In future work, we will incorporate more fine-grained and semantically rich graph representations to further improve the reasoning capabilities of MLLMs, especially in scenarios that require deeper relational understanding. This includes exploring the integration of knowledge graphs to provide structured world knowledge, as well as dynamic graphs that can evolve over time to reflect temporal changes or action progressions. Such extensions have the potential to significantly enhance MLLMs’ performance on more complex reasoning tasks and broaden their applicability across diverse domains.

---

### Author Response · Authors · 2025-12-01
**A Summary of Our Contributions and Revisions**

Dear Reviewers and ACs,

We thank the reviewers for taking the time to read our work and for their fruitful questions and comments. We truly believe that they have helped to strengthen the paper. We are delighted to see that the paper’s quality has been appreciated by all reviewers.

In this global response, we aim to clarify the contributions of our work and summarize the list of improvements we have made to the submission.

**Contributions:**

- **[Effective model]** We identify that existing MLLMs possess strong semantic understanding capabilities but fail to achieve precise spatiotemporal reasoning. To capture spatio-temporal information from videos, we develop a graph-based reinforcement learning framework for MLLM post-training, which is considered to **make sense** (5SFE), **reasonable** (KoMw), **interesting**, **novel** (hMWm), and **easy to follow** (q9jw).

- **[Pioneering dataset]** We construct an innovative and useful (hMWm) dataset to support the training of our proposed method, which is carefully constructed and **well aligned** (KoMw) with the task.

- **[Comprehensive experiments]** Video-STR is supported by **comprehensive** (KoMw) experiments, **promising** (5SFE, hMWm) comparison with prior work and **solid** (KoMw) ablation study.

**Responses and Revisions**

- **[For Reviewer KoMw]**
   - We further discuss the characteristics of the constructed dataset, the strengths and limitations of our graph-based method, and potential directions for future research.

- **[For Reviewer 5SFE]**
  - We elucidate the model’s reasoning process and present detailed information on data generation and filtering.
  - We add the ablation study of different reward functions.
  - We report the actual time and memory cost during the training.

- **[For Reviewer hMWm]**

   - We emphasize that our approach is mainly concerned with accurate spatial understanding of the physical world, rather than pixel-level information in digital images and videos.

   - We provide further detail on dataset construction, as well as on how the model is prompted.

   - We correct reference errors and improve the paper structure.

- **[For Reviewer q9jw]**

   - We add an experiment to validate that our method does not degrade the model’s original reasoning abilities, that is, its performance on general video understanding.

  - We highlight the distinct aspects of our model compared to Visual Genome, especially from the perspectives of motivation and methodology.

  - We clarify the application scenarios of the model, as well as its improvements on downstream tasks.

All modifications have been highlighted in blue in our revised manuscript. During the discussion, Reviewers hMWm and q9jw have already responded, and we believe that q9jw expressed an optimistic attitude. Thanks again for your efforts in reviewing our work, and we hope our responses can address any concerns about this work.

The Authors of Submission 7938.

---

### Meta-Review · Area_Chair_LWKZ · 2026-01-10

**Summary:**

The reviews for this work are divided into two groups. **Reviewer KoMw** supports the paper, but **Reviewers 5SFE, hMWm, and q9jw** remain critical for various reasons.

The authors propose using a relation graph with reinforcement learning for video reasoning and introduce the STV-205K dataset. While this setup yields results on some benchmarks, the critical reviewers identify recurring weaknesses. They highlight issues with methodological clarity and novelty. There are also concerns about the reliability of the auto-generated data and the narrow focus on object-centric tasks rather than general video understanding.

In the rebuttal, the authors clarified that the model handles object detection and tracking internally. They also explained the use of paraphrase templates and normalized bounding box centers. These details help, but they do not resolve the concerns regarding transparency, dataset reliability, and the limited scope of the contribution. The paper falls short of the bar for acceptance.

**Reviewer Concerns:**

**Reviewer KoMw** is generally positive about the work. Their main reservations involve the nature of the STV-205K dataset and the expressiveness of the relation graph.

They note that the dataset seems dominated by recognition-style questions instead of demanding reasoning tasks. They also suggest that the relation graph could model motion intentions rather than just spatial relations.

The rebuttal explains that the graph uses normalized bounding box centers and mentions prompts in the appendix. While these points clarify the implementation, they do not address whether the design captures higher-level reasoning. Since this reviewer is already positive and the empirical gains are present, these issues are viewed as extensions for future work.

**Reviewer 5SFE** raises concrete concerns that remain unresolved.

They requested key details on how the model predicts graph coordinates and how the authors ablated reward components. They also asked for the training cost of the GRPO process. The reviewer is concerned about systematic biases in the auto-generated QA dataset because it lacks human verification.

The authors clarified that the model handles detection internally and uses paraphrase templates to reduce bias. However, the prediction format for the graph remains vague. The authors did not provide the requested reward ablations or quantify the training time and memory costs. The concerns about transparency and the cost-benefit tradeoff of the approach are still open.

**Reviewer hMWm** is skeptical of the conceptual advance and the clarity of the paper.

They argue the method is a combination of existing techniques. The exposition is confusing, and several technical elements are poorly defined. Specifically, they point to missing information regarding Euclidean distance computations, the variable \(X\) in Equation 8, and the missing proof of Theorem 2. They also find the description of the TAO, ScanNet, and KITTI modalities to be ambiguous.

The rebuttal promises improved writing and clarifies the distance calculations. However, the fundamental issues remain. The theoretical support is incomplete, and the authors did not convincingly separate the effects of the method from the effects of the new dataset. The skepticism regarding novelty and clarity persists.

**Reviewer q9jw** focuses on the conceptual impact and the breadth of the work.

They argue the method is tuned for object-centric tasks and fails to improve general video understanding. The performance gap on Video-MME compared to state-of-the-art models supports this critique. The reviewer sees the work as an engineering effort and notes a lack of connection to prior graph-based grounding research.

The authors explicitly stated in the rebuttal that the method targets spatio-temporal reasoning rather than broad video understanding. While this is an honest clarification, it does not address the underlying concern about the narrow scope. There is still no discussion of the limitations of the object-centric design. The contribution appears primarily engineering-driven.

**Reviewer Scores:**

**Reviewer KoMw (Original: 8 → Predicted: 8)**
This reviewer sees the approach as a reasonable way to handle complex videos. They appreciate the alignment between the dataset and the framework. The rebuttal clarifies graph construction, which is consistent with the reviewer’s positive reading. The remaining concerns are about future directions, so the score will likely stay at 8.

**Reviewer 5SFE (Original: 4 → Predicted: 4)**
The score reflects concerns about implementation details and training costs. The rebuttal provides high-level answers but lacks the detailed evidence the reviewer requested. The absence of reward ablations is a significant gap. I see no reason for an upward revision.

**Reviewer hMWm (Original: 4 → Predicted: 4)**
The reviewer questions both novelty and clarity. The rebuttal clarifies some technical points but does not fix the conceptual gaps or the missing theoretical proof. Core concerns about the contribution remain, so the score should stay at 4.

**Reviewer q9jw (Original: 4 → Predicted: 4)**
The concern is the narrow focus and the lack of improvement on general benchmarks. The authors acknowledged the gap on Video-MME but did not change the assessment of the paper's impact. Without a stronger connection to prior work, the score is expected to remain at 4.

---

### Decision · Program_Chairs · 2026-01-26

Reject